# Does Health Literacy Reinforce Disease Knowledge Gain? A Prospective Observational Study of Hungarian COPD Patients

**DOI:** 10.3390/jcm10173990

**Published:** 2021-09-03

**Authors:** Orsolya Papp-Zipernovszky, Márta Csabai, Peter J. Schulz, János T. Varga

**Affiliations:** 1Institute of Psychology, University of Szeged, 6722 Szeged, Hungary; marta.csabai@psy.u-szeged.hu; 2Institute of Communication and Health, Università della Svizzera Italiana, 6900 Lugano, Switzerland; peter.schulz@usi.ch; 3Department of Pulmonology, Semmelweis University, 1083 Budapest, Hungary; janosvargaster@gmail.com

**Keywords:** health literacy, self-management, lifestyles, health services utilization

## Abstract

We set out to measure the health literacy (HL) of COPD patients using the Short Test of Functional Health Literacy (S-TOFHLA), perception-based screening questions (BHLS), and a specific disease knowledge test (COPD-Q). Our main focus is the relationship between functional HL and patients’ disease-knowledge, which contributes to the clarification of the relationship between the different kinds of HL. In two prospective observational studies, 151 COPD patients (80 males, mean age: 62 ± 9 years) completed a questionnaire containing HL measurements, psychological tests (e.g., the Self-Control and Self-Management Scale), and questions regarding subjective health status. Medical data of the patients from the MedSole system were added. The HL scores of the COPD patients were compared to a representative sample using a *t*-test. Furthermore, correlations of HL with demographic, psychological, and medical variables were calculated within the patient group. The relations among the different HL measurements were tested by chi-square trials. COPD patients had significantly lower HL, as measured by S-TOFHLA. Younger and higher educated patients possessed higher S-TOFHLA scores. Unlike the demographic variables, general self-management showed significant correlations with both BHLS and with COPD-Q. Out of the medical variables, objective health status was associated with BHLS and COPD-Q. Neither BHLS nor S-TOFHLA had a correlation with COPD-Q, but they correlated with each other. We found S-TOFHLA to be a better tool in the medical context. There is a clear gap between self-perceived/functional HL and the necessary disease knowledge. Rehabilitation care for patients with lower HL was more advantageous.

## 1. Introduction

A report by the United States Institute of Medicine identified health literacy (HL), which consists of diverse skills (e.g., reading, counting, communication, critical thinking), to be one of the most important means to improve population health [1]. By enabling or facilitating the access to and processing and application of health-related information, it can be assumed that this will impact treatment outcomes as well. For instance, health literacy was found to be negatively related to hospitalization [2] and positively related to health behaviors, such as medication adherence [3,4] or self-management, in a variety of chronic diseases [5].

The first empirical studies on the Hungarian validation of HL measurements were published in 2016 [6,7], and still, not much is known about its role in coping with chronic diseases. We know, however, that in a close-to-representative Hungarian sample, chronic patients showed lower HL than non-patients [6]. We conducted two studies between September 2016 and March 2018, pursuing the aim to further investigate HL in a sample of chronic inpatients and to identify their sociodemographic and psychological determinants as well as their biological and health behavior consequences. Our main focus is the relationship between functional health literacy and patients’ specific knowledge about their disease. We chose chronic obstructive pulmonary disease (COPD), the fourth most frequent cause of premature death, both in the world and in Hungary [8,9]. We are not aware of any Hungarian data on the impact of HL in terms of COPD patients and their knowledge about their disease on their health status.

### 1.1. Significance of Information, Education, and Lifestyle in COPD Management

COPD is a progressive dysfunction of the lung that restricts the patients’ activities, daily routine, and physical condition. It shows high comorbidity, with various cardiovascular and gastrointestinal diseases as well as osteoporosis and lung cancer [10]. There is also an increased risk of depression and anxiety disorders in COPD patients [11], which, together with hypoxemic states, has an effect on cognition [12], health literacy [13], quality of life, and even survival. Therefore, information, self-management, and education about the disease itself have become more important than they used to be.

Patients need to understand the pathophysiology and progression of the disease and how they can manage a complex therapeutic regimen. Previous studies have reported that medication adherence can be less than 50% [11], e.g., patients are likely to under-use maintenance therapy, and symptom-relieving drugs are often overused. Furthermore, the use of multiple devices, inadequate instruction, and low health literacy were found to be risk factors for high rates of inhaler misuse [14].

Besides adequate usage of inhalers, learning controlled breathing techniques and chest mobilization is necessary to improve lung mechanics and reduce chest hyperinflation. Patients also need information on the exacerbation of the disease and how to prevent it because it is strongly correlated with quality of life and survival. 

In addition to the specific techniques related to breathing, changes in lifestyle are also advised for COPD patients: smoking cessation, increased physical activity, and a change in diet [15]. Patients also need to learn how to cope with the increased presence of depression and anxiety.

### 1.2. HL in COPD: Determinants and Consequences

Despite its prevalence, COPD and, more specifically, its relationship with health literacy are still under-researched, especially in comparison to other diseases (e.g., diabetes). In the literature, there is no clear picture of the rate of inadequate health literacy among COPD patients: it is estimated between 13% ([16], an American asthma clinic) and 59% [17]. Older age and lower educational levels are important predictors of HL among COPD patients as well as patients in general [17].

From a clinical point of view, health literacy is associated with important patient outcomes. A good example comes from the study of Omachi et al. [18], who found that independently of the patients’ socioeconomic status, poor health literacy was associated with higher COPD severity, higher COPD helplessness, worse respiratory-specific quality of life, and higher odds of COPD-related emergency healthcare utilization, with a lower degree of self-management. In a follow-up study by Puente-Maestu et al. [17], health literacy was an independent predictor of mortality. Individuals with inadequate HL reported lower quality of life and health status than patients with an adequate level of HL, while their lung function was no different. Those with inadequate HL also had a higher risk of having two comorbidities, need of assistance, anxiety/depression, admissions or visits to the emergency department, and all-cause deaths in the following year.

Knowledge of the patients’ own illness is considered both a type of HL (see declarative knowledge in Schulz and Nakamoto’s model [19]) as well as a patient outcome [20]. DeWalt et al. [21] conducted a meta-analysis of articles on patients suffering from chronic diseases. Patients who performed worse on a health literacy test knew less about their health, the healthcare system, therapies, and their own chronic illness. Williams et al. [16] and Gazmararian et al. [22] confirmed these results for asthma patients and Kale et al. [5] and Puente-Maestu et al. [17] for COPD patients. In our assumption, knowledge about an illness can be a mediating variable between functional HL and patient outcomes, which might give an explanation of how HL influences the improvement of health status.

Whether adequate (F)HL is a prerequisite of self-management in chronic disease [23,24,25,26] is still a debated question, but self-management—the process of actively engaging in self-care activities, with the goal of improving one’s behavior and well-being ([27] p. 56)—is considered to be an important pillar in the health outcomes of chronically ill patients. A third overlapping theoretical construct is patient empowerment (PE), which, from an intrapersonal perspective, refers to the patient’s perceived capacity to participate in treatment-related decision-making [28]. Increasing PE has become a widely used tool to improve health outcomes in chronic conditions [29]. Schulz and Nakamoto [28] adapted the operationalization of PE from management literature [30], where it contained four dimensions: Meaningfulness (the belief that investing energy in a certain action pays off); Competence (the belief in one’s own ability to implement certain actions); Impact (the belief that one’s actions can affect certain outcomes); and Self-determination (one’s self-induced motivation to implement an action) ([31] p. 511). Recent models (e.g., Wanga et al. [32]) handle at least two of these variables to determine the necessary elements of better health outcomes and more effective patient education. In our paper, we test the assumptions of the Health Empowerment Model (HEM) [28], which claims HL and PE to be independent constructs that determine health outcomes, e.g., subjective health status or the frequency of using the healthcare system, in interaction.

### 1.3. Aims and Hypotheses

The specific aim of our two successive observational studies is to describe the health literacy level of COPD patients in Hungary, to compare it to national health literacy levels, and to identify its determinants and associations in a large Hungarian inpatient COPD sample. Our specific focus is the association between different indicators of health literacy: self-reported difficulties in filling out forms and reading hospital materials, reading skills, and specific knowledge of COPD. We formulated our hypotheses along these aims:

**Hypothesis** **1** **(H1).**
*Hungarian COPD patients will show a lower level of HL than the population at large (see the results of other studies with COPD patients [14,17,33]).*


**Hypothesis** **2** **(H2).**
*HL correlates positively with psychological variables that indicate better self-management, as was also found in the research conducted by Omachi et al. [18]. The HEM [28] questions this assumption.*


**Hypothesis** **3** **(H3).**
*Different indicators of HL (measured by BHLS, S-TOFHLA, and COPD-Q) will be associated with each other. Only a few studies have used more than one HL measurement, especially in the context of respiratory diseases. Results indicate that functional and self-reported HL are positively associated with each other [34] and with specific knowledge about the patients’ illness [5,16,17,22].*


**Hypothesis** **4** **(H4).**
*HL will be associated with biological (lung function test) and behavioral (subjective health status, COPD Assessment Test, six-minute walking test, and the frequency of using the healthcare system) patient outcome measurements. One of the main reasons underlying the popularity of the construct of HL is its associations with health outcomes [2], up to the negative association between HL and mortality. In contrast, Puente-Maestu et al. [17] found no association between HL and lung function.*


## 2. Materials and Methods

We used a cross-sectional quantitative research design in both studies because validated questionnaires in Hungarian were available and we wanted to reach a large group of COPD patients to get a picture of their HL levels.

### 2.1. Study 1

#### 2.1.1. Participants and Procedure

The data collection was carried out between September 2016 and July 2017 at the inpatient Department of Rehabilitation of the National Koranyi Institute for Pulmonology. Inclusion criteria were having a COPD diagnosis and not having a severe vision problem that could not be corrected with glasses. With a convenience sampling, 80 patients (37 females, 43 males; with a mean age of 62 ± 9) were enrolled to fill in HL questionnaires (S-TOFHLA and BHLS), an empowerment questionnaire (HES), and demographic data. After receiving information about the aim and process of the study, putative participants were asked to sign an informed consent to participate. Six subjects preferred not to participate in the study, referring to fatigue or lack of time (waiting for a medical examination). The Institute has its own ethical board, which gave the permission to carry out the study. The study was registered at the ISRCTN registry under ISRCTN13019180 ID. 

#### 2.1.2. Measures

##### Personal Characteristics

Sex, age, marital status, education, profession, and income were self-reported by the participants.

##### Health Literacy Measures

The reading comprehension section of the Short Test of Functional Health Literacy (S-TOFHLA [23], Hungarian version [6]) was used to measure health literacy. S-TOFHLA is a performance-based test, which consists of a reading comprehension section and a numeracy section. We decided to include only the reading comprehension section, as the numeracy scores had a low internal consistency in the Hungarian context. The reading comprehension section has two passages (the first is about upper gastrointestinal screening and the second about the patients’ rights and responsibilities), adding up to 36 cloze test items with multiple-choice options. The respondent has to choose from four answer options of meaning-carrying words, which are left blank. The former passage is equal to a 4th-grade reading comprehension level (age 10); the latter corresponds to a 10th-grade level (age 16). The reading comprehension section has to be filled in by the respondents and has a 7-min time limit: the respondents overrunning the limit are stopped, or their answers are not registered. A higher score on the test means a higher level of HL.The Brief Health Literacy Screen (BHLS) is a self-report questionnaire consisting of 3 questions [34] (Hungarian version [6]). This tool provides a rapid way to detect patients with inadequate or marginal health literacy; therefore, it is especially appropriate for a clinical setting. Subjects need to answer questions such as “How often do you have someone help you read hospital materials? ” on a 5-point scale, where “0” is “Never”, while “4” is “Always”. A higher score in the test indicates lower HL.

##### Psychological Tests

The 12-item Health Empowerment Scale (HES) was used to measure patient empowerment [28]. The scale was adapted from the Psychological Empowerment Questionnaire [30] to the health context. The tool consists of four 3-item factors to measure all the subfacets of empowerment: Competence (“I am confident about my ability to deal with my health”), Meaningfulness (“Dealing with my health is very important for me”), Impact (“I have a great deal of control over managing my health”), and Self-determination (“I can decide on my own how to handle my health”). Each item has to be answered on a 7-point Likert scale, which ranges from 1 (I do not agree at all) to 7 (I completely agree). Higher scores in both the subscales and the summarized result mean more empowered patients. 

##### Patient Outcomes: Subjective Health Status

In Study 1, we only measured a self-rated patient outcome, i.e., the participants’ subjective health status. In a single question, we asked them to provide an assessment regarding their current health status on a 5-step scale, ranging from 1 (very bad) to 5 (excellent).

### 2.2. Study 2

After having analyzed the data of Study 1, we retained our aims and hypotheses and our observational research design, but we added the illness-specific HL questionnaire COPD-Q [35], new psychological measurements (Mini Mental State Examination (MMSE), Self-Control and Self-Management Scale (SCMS)), and biological as well as behavioral data as outcome measures that were available in the MedSole system. 

#### 2.2.1. Participants and Procedure

Data collection with the partly new questionnaires was run between January and March 2018 at the same Institute, with a slightly modified procedure: all patients who had COPD diagnoses were visited in their wards. The inclusion criteria were widened to include not having medium or severe cognitive impairment. (We used a performance-based cognitive test to measure cognitive impairment. The Mini Mental State Examination [36] is a 30-point questionnaire that includes tests of orientation, attention, memory, language, and visual–spatial skills. It is used extensively in clinical and research settings. Performance on the MMSE has also been positively correlated with functional health literacy [37]. The fixed cutting point for exclusion was under 13 points. No subjects needed to be excluded based on the MMSE score.) In total, 71 patients (39 females, 32 males; with a mean age of 64 ± 7; suffering from the disease for 7.8 ± 7.6 years) fulfilled the test battery after signing the informed consent; 6 subjects rejected participation in the study. The Institute confirmed the modification of our study and the second run of data collection.

#### 2.2.2. Measures

##### Personal Characteristics

We added the number of chronic illnesses and duration of COPD to the demographic data collected in Study 1.

##### Health Literacy Measures

In clinical HL studies, illness-specific questionnaires are frequently used either as declarative HL or a knowledge-based outcome of HL. Besides S-TOFHLA and BHLS in Study 2, we used the Chronic Obstructive Pulmonary Disease Knowledge Questionnaire (COPD-Q, [35]), which is a 13-item instrument designed to measure knowledge of COPD on a 5th-grade educational level, e.g., “People with COPD should have a flu shot every year”. Patients can answer each question by marking the “True”, “False”, or “Not sure” possibility. Higher scores indicate better knowledge about COPD. There is no Hungarian validated version of this instrument; therefore, we made a translation following the requirements of adapting psychological tests [38]: two independent translators made two Hungarian versions, which were compared and discussed by an expert group containing a pulmonologist and health psychologists, who finalized a Hungarian translation. 

##### Psychological Tests

In Study 1, we used a rather complex construct, health empowerment to measure the meaning and control of health, as felt by the patients. In Study 2, we aimed at using a more direct measurement of self-management. The 16-item Self-Control and Self-Management Scale (SCMS [39]) is an adult general trait measure of self-management, where subjects need to indicate their agreement with statements such as “When I am working toward something, it gets all of my attention”(self-monitoring scale), “The goals I achieve do not mean much to me”(self-evaluating scale), or “I congratulate myself when I make some progress” (self-reinforcing scale) on a six-point scale (0—Very undescriptive of me, 1—Mostly undescriptive of me, 2—A little undescriptive of me, 3—A little descriptive of me, 4—Mostly descriptive of me, 5—Very descriptive of me). Higher scores mean higher self-control and self-management. The instrument is currently under validation in Hungarian.

##### Patient Outcomes: Current Objective (CAT™) and Subjective Health Status in COPD and the Frequency of Health System Use

Besides the self-evaluated current health status in Study 2, we aimed at using more objective variables: The COPD Assessment Test (CAT™, [40]) is an indicator of COPD patients’ objective health status and is available in the MedSol system. CAT^TM^ assesses the symptoms experienced by people with COPD, such as ongoing cough, shortness of breath, wheezing, and chest tightness, which can restrict them from performing simple activities such as washing or getting dressed. It is designed to measure the impact of COPD on a person’s life and how these change over time. It contains 8 items, with a total score of 0–40. Higher scores indicate stronger symptoms and a higher impact on everyday life. CAT™ was completed twice during the patients’ stay in the hospital: at arrival (CAT1, see in Table 1) and before leaving (CAT2). The change between the two states was also calculated (CAT_diff).

Another “more objective” indicator can be the frequency of using the healthcare system. We asked the subjects to indicate how often (0, 1, 2, 3, 4, 5–9, 10 or more times) in the last 12 months they had visited a health professional, an accident and emergency clinic, or a hospital department as an inpatient due to their own COPD condition. This question was adapted from the survey of Schulz et al. [41], who developed it to identify the relation between e-health literacy and the use of healthcare systems.

##### Functional Tests

After hospitalization, all patients underwent a lung function test by spirometry (Piston PDD-301/sh, Budapest, Hungary) to determine the degree of airway obstruction based on American Thoracic Society/European Respiratory Society guidelines [42]. All of the COPD patients inhaled 400 mg of salbutamol before testing. We determined FEV1 (forced exhaled volume in the first second), FVC (forced vital capacity), and airway obstruction (FEV1/FVC ratio) in the test.

The six-minute walking test (6MWT) is a field test to measure the exercise capacity of patients [43]. It lasts 6 min. The patients were instructed to walk as fast as possible, and they were controlled throughout the test. Oxygen saturation and heart rate were measured, and the Borg scale was evaluated by modified Borg score before, during, and after the test.

### 2.3. Statistical Analysis

Statistical analysis of the data was performed using IBM SPSS for Windows 22 (IBM Corporation, Armonk, NY, USA). According to Kolmogorov–Smirnov tests, conditions for normality were not always met (see Table 1 in the Results section); therefore, we decided to use non-parametric tests in the statistical analyses of both studies. This was also consistent with the smaller sample size of the individual studies as well as with some of our variables that included ordinal scales. Internal consistency and reliability of the measurements were tested with Cronbach’s alpha, and Spearman’s rank correlation was used to examine the associations between the variables. Differences between groups were compared by the Mann–Whitney U-test and Dunett’s T3 post hoc test. We also agreed that the statistically significant *p*-value should be <0.05.

## 3. Results

We summarized the descriptive characteristics of our variables resulting from the questionnaires in Table 1. It contains the normality and internal consistency values and the mean and standard deviation of the measurements in our sample in both studies. It shows S-TOFHLA to have the only acceptable internal consistency (Cronbach α = 0.94) among our HL measurements. We decided not to leave out the other tests, but we need to interpret their results with caution.

### 3.1. Study 1

To identify the level of health literacy of Hungarian COPD patients, the S-TOFHLA and BHLS were applied. For comparison, we used our previous results of a close-to-representative sample of the Hungarian population (see [6] or [31] for more details). The significance of differences was determined by one-sample *t*-tests. COPD patients had significantly lower HL than the standard value of both S-TOFHLA (t = −7.64, df = 79, *p* < 0.001) and BHLS (t = 7.426, df = 79, *p* < 0.001) measurements. Furthermore, in the representative Hungarian sample, 14.3% showed less than adequate HL in S-TOFHLA [6], while among COPD patients, 46.3% had inadequate or marginal health literacy (under the score of 23). These results confirmed H1, which claims that Hungarian COPD patients show a lower level of HL than the population at large.

Our second analysis referred to the associations of health literacy with age (negative) and educational level. Younger patients (rho = −0.438, df = 78, *p* < 0.01) and COPD patients with a high-school or university degree (t-education = −2.115, df = 2, *p* = 0.039) possessed higher HL based on their S-TOFHLA scores. However, educational group and age did not differentiate health literacy scores significantly in BHLS.

We used a rather complex variable, empowerment, which reflects patients’ commitment and sense of control to maintain their health as much as they are able to. We supposed that HL correlates positively with empowerment (H2). In contrast to our expectations, S-TOFHLA correlated negatively with the self-determination subscale of health empowerment (rho = −0.226, df = 78, *p* < 0.05). No other significant associations were found.

Our theoretically important focus is the association of the different types of health literacy skills (H3): whether functional HL, which is based on reading or numerical performance, predicts self-reported health literacy. Using a linear regression model, we found S-TOFHLA did not predict BHLS. This contradicts H3.

We assumed that HL determines subjective health status as a health outcome; however, using Spearman’s rank correlation, there was no significant association between either BHLS or S-TOFHLA and subjective health status. This result did not confirm H4.

### 3.2. Study 2

To identify the level of health literacy of Hungarian COPD patients, the S-TOFHLA and BHLS were applied again. Since we had data from Study 1 using the same tests, we ran a *t*-test to check whether there were significant differences between the patient groups. The differences were not significant (t_BHLS_ = −0.193, df = 149, *p* = 0.847; t_S-TOFHLA_ = −1.01, df = 149, *p* = 0.314), therefore we used the scores of all COPD patients. The significance of differences was determined by one-sample *t*-tests using the standard Hungarian means of the tests. COPD patients had significantly lower HL on S-TOFHLA than the standard value (t = −9.32, df = 148, *p* < 0.001), a difference which remained after we adjusted the trial for age (t = −2.481, df = 148, *p* < 0.02). Furthermore, in the representative Hungarian sample, 14.3% showed less than adequate HL in S-TOFHLA [6], while among COPD patients, 42.6% had inadequate or marginal health literacy; using a chi-square trial, this difference is significant (chi-square = 23.572, df = 1, *p* < 0.001). These results confirmed H1, that Hungarian COPD patients possess a lower level of HL than the population at large, while there was no difference in the BHLS measurement (t = −0.917, df = 149, *p* > 0.05) compared to the standard Hungarian value.

Our second question referred to the associations of health literacy in the Hungarian COPD sample (with demographic variables, with self-management (H2), and with patient outcomes (H4)). To identify determinants and consequences, we took the S-TOFHLA, BHLS, and COPD-Q as variables of health literacy and correlated them with age, self-management scores, biological (MWD difference, FEV%), and “health-outcome” variables (subjective and objective health status, frequency of using health services). We used Spearman’s rank correlation as a non-parametric test and the Mann–Whitney U-test, where educational groups were compared. Younger patients (r = −0.27, df = 149, *p* < 0.05) and higher educated COPD patients (F-education = 5.588, df = 3, *p* < 0.01; see Table 2) possessed higher HL based on their S-TOFHLA scores. The significant difference was between primary and secondary education according to the post hoc test. However, educational background (having primary school, vocational school, or secondary school education or university degree) and age did not differentiate health literacy scores significantly on either BHLS or COPD-Q.

S-TOFHLA showed a weak but significant correlation with the self-management evaluation subscale (rho = 0.276, df = 69, *p* < 0.05) (H2). Unlike demographic variables, general self-management skills also showed significant correlations with both BHLS (reinforcement subscale rho = −0.242, df = 69, *p* < 0.05) and COPD-Q (SMCSsum rho = 0.240, df = 69, *p* < 0.05) in the expected directions: higher HL skills mean higher ability to manage the disease. H2 is confirmed by our results: all health literacy measurements showed a low but significant correlation with self-management.

Only two out of the biological and health outcome variables showed associations with our health literacy tests in Study 2 (H4): BHLS correlated weakly and negatively with objective health status measured before leaving the hospital (CAT2) (rho = −0.303, df = 38, *p* = 0.057), and COPD-Q showed a weak, positive association with the change of objective health status during the patients’ stay in the hospital (CAT_diff) (rho = 0.332, df = 38, *p* < 0.05). The first result indicates that patients with lower self-reported health literacy report better health conditions at the end of their hospital treatment. The second finding means that patients with higher knowledge about their disease had lower remissions in the hospital. H4 was not confirmed.

Our specific question referred to the associations of the different types of health literacy skills (H3), or, to put it differently, whether functional or self-reported health literacy predicts patients’ knowledge about their disease. Neither BHLS nor S-TOFHLA had a correlation with COPD-Q, but in our combined COPD sample (Study 1 and Study 2), they correlated with each other in the expected direction (rho = −0.289, df = 149, *p* < 0.05). We also calculated chi-square trials to test whether a low literate person could have more or less knowledge about his/her illness, but no associations were found among these variables in Study 2 (chi-square S-TOFHLA and COPD-Q = 0.142, df = 69, *p* > 0.05 and chi-square BHLS and COPD-Q = 0.403, df = 69, *p* > 0.05). These findings mean that in our sample, no association can be found among self-reported HL, functional HL, and declarative knowledge of the illness, which contradicts H3.

## 4. General Discussion

Measuring HL in a clinical context is a new attempt in Hungary to develop more effective techniques in patient education and to provide more complex patient care. Our study is among the first steps into this direction when trying to focus on the HL level of a special patient group, i.e., persons with COPD, on its psychosocial determinants, and, most importantly, on the question of whether patients’ self-perception of their ability to interpret health-related texts is associated with more objective application and knowledge-based measurements of health and illness. We consider the examination of this patient group as a strength of our study because, despite its significance in mortality [8,9], it is still under-researched internationally.

In our first hypothesis, we assumed that the HL level of Hungarian COPD patients would be lower than that of the total population. Our results, with the combined sample of Study 1 and Study 2, support this assumption because the COPD patients had significantly lower HL on S-TOFHLA than the score representing the population at large. Furthermore, 43% of COPD patients showed less than adequate HL, which is in accordance with the literature (29–59%, [5,14,17]). This stresses a demand for further help and education of chronically ill people in Hungary to enable them to understand the nature of their illness as well as the healthcare system [6]. This result also highlights the importance of clinical HL studies: knowing health literacy and the differences between certain patient groups can help optimize patient education. It should be noted, however, that the other HL measurements we used did not show this difference between the population and the patient group. One possible explanation for this is the low internal consistency of these measurements in our sample (see Table 1), which is a limitation of our study that needs further examination.

Our results regarding age, education (with the combined sample of Study 1 and Study 2), and biological variables (H4) is in concordance with the literature [17]: younger and higher educated COPD patients scored higher on S-TOFHLA, showing better performance in applying their reading skills to the health context. This relation between lower level of HL and older age as well as lower education level is also well-known regarding the normal population (e.g., [36,44]). Again, self-reported HL levels and knowledge about the illness were not associated with age and educational level in our sample.

It is a strength of our study that we included more objective, behavioral (6MWT), and biological (lung function) data of the COPD patients. The lack of associations of these with our HL tests is in accordance with the literature [17], which points out the necessity of analyzing mediator variables between HL skills and health status, such as health beliefs or self-efficacy. CAT™ might take a borderline position between subjective and objective health status because it is administered by healthcare workers, and its score is based on the presence and strengths of the patients’ symptoms. Our findings show that a more objective health status is associated in an unexpected but interpretable direction with the self-reported health literacy (BHLS) scores: patients with lower HL reported better health conditions by the end of their stay in the hospital. Furthermore, there was an association between patients’ higher knowledge about their disease and their weaker capacity to change their health status during their hospital stay. A possible interpretation of that is that more knowledgeable patients do not need the hospital to control their health status better; their stay in the Institute is rather uncomfortable for them. In contrast, patients with lower HL cannot manage their health condition well at home, but they get professional support in the hospital, which increases their objective health status. These factors lead us to the assumption that increasing knowledge can result in more useful routines in illness management at patients’ homes. On top of this explanation, methodological reasons may also contribute to these results: the low internal consistency of BHLS and COPD-Q as well as the possible bias of a self-reported questionnaire such as BHLS.

In our second hypothesis, we formulated assumptions regarding the positive associations between HL, health-empowerment (Study 1), and a general trait measure of self-management (Study 2). According to our results, all health literacy measurements showed low but significant correlations with self-perceived self-management. This was also found in the COPD population by Omachi et al. [18]. However, the health empowerment scale correlated with functional health literacy in Study 1 in an unexpected (negative) direction and did not correlate with other types of HL (Study 1). These findings need further explanation: we refer here again to Schulz and Nakamoto’s health empowerment model [28], which claims that health literacy and psychological empowerment are separated but interrelated constructs in terms of determining health outcomes. They also proved their model in the case of asthma patients [17]. This can explain the lack of associations between BHLS and HES. The negative correlation between S-TOFHLA and HES leads us to a careful assumption regarding Hungarian COPD patients’ attitudes toward their health (HES measures patients’ competence, meaningfulness, impact, and self-determination in connection to their health): the ones who can use written information more effectively in understanding issues in health and illness feel less self-determined in maintaining and controlling their health status due to the overall impact of COPD.

The other strength of our study is that we used three different kinds of HL measurements, which provides the opportunity to test its associations (H3). One of our results with the combined sample confirms the low but significant relationship between functional and self-reported HL reported in the literature ([34]; with a close-to-representative Hungarian sample, see [6]). It means that those COPD patients who claim that they deal with formulas in the health context more effectively are the ones who perform better in reading health-related texts. The unexpected gap is between the self-reported and actual capacity of gaining information in health-related context and the patients’ knowledge about their own illness: neither the BHLS nor S-TOFHLA predicted the level of COPD-Q. It means that other factors determine who has the required knowledge of their illness. A similar inconsistency was found by Koltai and Kun [7] using HLS-EU and New Vital Sign scales on a representative Hungarian population: there was no association between the self-reported HL and the functional skills, which they interpreted as a hint that Hungarian people do not have an everyday routine in applying their skills. Our explanation focuses more on the motivational aspects: Hungarian COPD patients lose their motivation to become a good self-manager after the emergence of their diagnoses and do not invest energy into gaining specific knowledge of their disease even if they have an acceptable level of HL. New research should uncover whether this reflects a fatalistic attitude regarding health and illness or maladaptive coping strategies.

### The Limitations of Our Study

Previously, we highlighted some strengths of our study, but we also need to point out its weaknesses. We decided to use a cross-sectional study design to identify the main characteristics of Hungarian COPD populations’ HL. However, this design is not suitable for making causal inferences (e.g., between functional HL and illness knowledge); the identified associations might be difficult to interpret (e.g., the negative relation between BHLS and CAT2), and it is susceptible to certain biases (e.g., nonresponse bias) [45]. The collected data derives from one (although national) center of Pulmonology in Hungary. We used two sets of questionnaires in two rounds of data collection, which resulted in only 71 patients who fulfilled COPD-Q. To draw valid conclusions about the knowledge of COPD patients on their illness, more participants are needed to fill in this survey. Some of our measurements are self-reported questionnaires (e.g., BHLS, subjective health status, HES), which can be a source of response bias, such as social desirability. The internal consistency of BHLS and COPD-Q was rather low in our sample (see Table 1); therefore, we need to interpret our results with caution.

## 5. Conclusions

Our findings suggest S-TOFHLA is the most appropriate tool to measure patients’ HL because it had the only acceptable (0.94) reliability in our sample (see Table 1) and it showed the expected correlations with patients’ demographic data. Our findings reveal a kind of inconsistency in the Hungarian COPD patients’ sample between the self-perception of their ability to understand health-related information and their knowledge about their illness. On the one hand, it means that they need further help in acquiring appropriate, personalized information about their illness, which, at the same time, motivates them to maintain daily self-care activities. On the other hand, our findings suggest that improving their knowledge about a severe illness, such as COPD, can also be threatening for patients without developing their self-management and coping skills in the meantime. However, introducing COPD-Q in pulmonology rehabilitation departments might reveal topics that are complicated for the patients and may help clinicians to provide HL interventions with appropriate instructions and educational materials, aiming to improve the health outcome of those living with COPD. To measure HL before and after patient educational programs is a potential future research option, which may uncover in real-life context the determinants of gaining knowledge in the Hungarian COPD population.

## Figures and Tables

**Table 1 jcm-10-03990-t001:** Normality, internal consistency, and descriptive values (mean ± SD) of our measurements in Study 1 (*n* = 80), Study 2 (*n* = 71), and in sum (*n* = 151).

Measurement/ Descriptive Statistics	S-TOFHLA Reading Part	BHLS	COPD-Q	SCMS	HES	CAT1
Normality						
Study 1 (*n* = 80)	0.12 **	0.11 *			0.14 **	
Study 2 (*n* = 71)			0.21 **	0.1		0.1
Sum (*n* = 151)	0.17	0.09 *				
Internal consistency						-
Study 1 (*n* = 80)	0.946	0.57			0.886	
Study 2 (*n* = 71)			0.51	0.71		
Sum (*n* = 151)	0.94	0.46				
Mean ± SD						
Study 1 (*n* = 80)	23.11 ± 1.02	6.73 ± 0.33			69.3 ± 1.2	
Study 2 (*n* = 71)			7.55 ± 2.24	54.9 ± 9.89		17.88 ± 8.15
Sum (*n* = 151)	23.89 ± 8.9	4.08 ± 2.37				
Standard data in the literature	30.64 ± 7.67	4.25 ± 2.5	7.58 ± 2.93	-	-	-
			t = −0.115			

S-TOFHLA = Short Test of Functional Health Literacy, BHLS = Brief Health Literacy Screen, COPD-Q = Chronic Obstructive Pulmonary Disease Knowledge Questionnaire, SCMS = Self-Control and Self-Management Scale, HES = Health Empowerment Scale, CAT™ = COPD Assessment Test; * *p* < 0.05, ** *p* < 0.01.

**Table 2 jcm-10-03990-t002:** Mean and standard deviation values of the HL tests according to educational background.

HL-Test/ Educational Level	*N*	BHLS		S-TOFHLA_Reading		*N*	COPD-Q	
		Mean	SD	Mean	SD		Mean	SD
1 (primary school)	40	4.68 ± 2.068		20.125 ± 9.14		11	7.27 ± 3.47	
2 (vocational school)	53	4.23 ± 2.785		23.13 ± 8.15		19	8.05 ± 1.96	
3 (secondary school)	39	3.82 ± 2.405		27.49 ± 8.28		22	7.36 ± 1.81	
4 (university degree)	19	4.53 ± 2.144		25.84 ± 7.36		19	7.42 ± 2.19	

S-TOFHLA = Short Test of Functional Health Literacy, BHLS = Brief Health Literacy Screen, COPD-Q = Chronic Obstructive Pulmonary Disease Knowledge Questionnaire.

## Data Availability

The study materials and the detail of all analyses are available from the corresponding author upon request.

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
