# Peer review of "Does Health Literacy Reinforce Disease Knowledge Gain? A Prospective Observational Study of Hungarian COPD Patients"

_jcm, 2021, doi:10.3390/jcm10173990_

Round 1

Reviewer 1 Report

The article has improved a lot since the last version. The authors are to be congratulated for the effort. However, a few minor changes are needed before it can be published.

  These changes are necessary to improve the quality and readability of the article.

  1. The authors indicate in lines 58-60 that depression and anxiety, and hypoxemia affect health literacy. For that, they rely on reference 14 (Omachi, T.A.; Sarkar, U.; Yelin, E.H.; Blanc, P.D.; Katz, P. Lower Health Literacy is Associated with Poorer Health Status and 608 Outcomes in Chronic Obstructive Pulmonary Disease. Journal of General Internal Medicine 2012, 28(1), 74-81.) This is not true because reference 14 specifies that patients with poor health literacy have a greater COPD severity. It does not say that COPD influences health literacy but that health literacy influences COPD outcomes. Moreover, this contradicts what is stated in the same article in the paragraph from lines 91 to 101, indicating that illiteracy influences poor COPD outcomes, not the other way around. Lines 58 to 60 should be rewritten or even deleted because they deal with the same aspect that is reiterated later in paragraphs 91 to 101.

  1. Statistical information should be presented in the usual presentation format for statistical information in medical journals. The format is as follows. (chi-square = xxx, df= yyy P= zzz) In their response, the authors have clarified the statistical information they present. Statistical information is presented following the APA Style Guide 7th edition. The APA (American Psychological Association) guide is used primarily in psychology and social science publications but not in medical journals. Readers of medical journals are not familiar with this form of presenting statistical information that is common in psychology journals. Giving statistical information in APA format make it difficult for them to understand the article.

Author Response

I would like to thank you for devoting your expertise and time to imrove our paper. Please, find our answers below:

  1. The authors indicate in lines 58-60 that depression and anxiety, and hypoxemia affect health literacy. For that, they rely on reference 14 (Omachi, T.A.; Sarkar, U.; Yelin, E.H.; Blanc, P.D.; Katz, P. Lower Health Literacy is Associated with Poorer Health Status and 608 Outcomes in Chronic Obstructive Pulmonary Disease. Journal of General Internal Medicine 2012, 28(1), 74-81.) This is not true because reference 14 specifies that patients with poor health literacy have a greater COPD severity. It does not say that COPD influences health literacy but that health literacy influences COPD outcomes. Moreover, this contradicts what is stated in the same article in the paragraph from lines 91 to 101, indicating that illiteracy influences poor COPD outcomes, not the other way around. Lines 58 to 60 should be rewritten or even deleted because they deal with the same aspect that is reiterated later in paragraphs 91 to 101.

REPLY1: We apologize for the imprecise reference. We deleted Reference14 from that sentence and copy here the relevant statement from Reference13 (Sadeghi, S.; Brooks, D.; Stagg-Peterson, S.; Goldstein, R. Growing Awareness of the Importance of Health Literacy in Individuals with COPD. COPD: Journal of Chronic Obstructive Pulmonary Disease 2013, 10(1), 72-78), which we used as the basis of our claim: “In patients with COPD, diminished health literacy may be compounded by impairments in abstract reasoning and memory associated with hypoxemia (16) as well as by secondary impairments of depression and anxiety, which influence learning, comprehension and decisionmaking ability (17).” (Sadeghi et al., 2013, 73.)

If it is still required we can modify our sentence to: "There is also an increased risk of depression and anxiety disorders in COPD patients [11], which might have an effect on cognition [12], health literacy [13], quality of life and even survival."

2. Statistical information should be presented in the usual presentation format for statistical information in medical journals. The format is as follows. (chi-square = xxx, df= yyy P= zzz) In their response, the authors have clarified the statistical information they present. Statistical information is presented following the APA Style Guide 7th edition. The APA (American Psychological Association) guide is used primarily in psychology and social science publications but not in medical journals. Readers of medical journals are not familiar with this form of presenting statistical information that is common in psychology journals. Giving statistical information in APA format make it difficult for them to understand the article.

REPLY2: The presentation of statistical information is corrected according to medical journals throughout the manuscript.

Reviewer 2 Report

Overall, the manuscript is interesting, well-written, and novel. The authors measure health literacy in COPD patients using validated questionnaires. They use correlation tests to measure associations between health literacy and responses from a self-management scale and health status. The findings are presented from two studies and discussed. 

Introduction:

The introduction is well-researched, organized, and flows well. The objectives of the paper are presented clearly and the authors ask well thought out research questions. 

Line 134: "researches" does not fit here. Use "studies" instead. 

Methods: 

The methods are thorough and the validation of each questionnaire used is presented clearly. The statistical data analysis and tests performed were appropriate for the presented methods and explained succinctly. 

Lines 198, 202 and throughout the methods: Sometimes there is a space after "Study 1" and sometimes there is no space "Study1". Please check this throughout the manuscript to ensure consistency. I recommend leaving a space after "Study 1". 

Results:

The results are organized and present well. 

Discussion:

The discussion section is organized well and the strengths highlighted are important. However, I would like to see the limitations section expanded. What limitations exist using self-reported questionnaires, cross-sectional study design, etc? 

Additionally, what are other possible reasons for the negative correlation between health literacy and health outcome? Could the methods be contributing to this? Are there other possible explanations? 

Conclusion: 

I would like to see more in-depth conclusions drawn based on the results obtained. Please also add more information about potential future research. 

Author Response

We would like to thank you for devoting your expertise and time to improve our paper. Please, find our replies below:

1. Line 134: "researches" does not fit here. Use "studies" instead. 

REPLY1: Corrected accordingly.

2. Lines 198, 202 and throughout the methods: Sometimes there is a space after "Study 1" and sometimes there is no space "Study1". Please check this throughout the manuscript to ensure consistency. I recommend leaving a space after "Study 1". 

REPLY2: We left a space after „Study 1” and „Study 2”, and corrected throughout the manuscript consistently.

3. Discussion: I would like to see the limitations section expanded. What limitations exist using self-reported questionnaires, cross-sectional study design, etc? 

REPLY3: We added the following to the Limitations section:

„We decided to use a cross-sectional study design to identify the main characteristics of Hungarian COPD populations’ HL. However, this design is not appropriate for making causal inferences (e.g. between functional HL and illness-knowledge), the identified associations might be difficult to interpret (e.g. the negative relation between BHLS and CAT2) and it is susceptible to certain biases (e.g. nonresponse bias) [45].”

„Some of our measurements are self-reported questionnaires (e.g. BHLS, subjective health status, HES), which can be a source of response bias, such as social desirability.”

4. Additionally, what are other possible reasons for the negative correlation between health literacy and health outcome? Could the methods be contributing to this? Are there other possible explanations? 

REPLY4: We added the following to the General Discussion part after explaining the mentioned negative results from line457: “On the top of this explanation, methodological reasons may also contribute to these results: the low internal consistency of BHLS and COPD-Q as well as the possible bias of a self-reported questionnaire like BHLS.”

5. Conclusion: I would like to see more in-depth conclusions drawn based on the results obtained.

REPLY5: We added the following to Conclusions: „On the one hand it inspires that they need further help in acquiring the appropriate, personalized information about their illness, which at the same time motivates them to maintain daily self-care activities. On the other hand, our findings suggest that improving the knowledge about a severe illness, such as COPD can also be threatening for patients without developing their self-management and coping skills in the meantime.”

6. Please also add more information about potential future research. 

REPLY6: We added the following to Conclusions: „To measure HL before and after patient educational programs is a potential future research option, which may uncover in real-life context the determinants of gaining knowledge in Hungarian COPD population.”

Round 2

Reviewer 2 Report

All of my concerns and comments were addressed. Thank you! 

This manuscript is a resubmission of an earlier submission. The following is a list of the peer review reports and author responses from that submission.

Round 1

Reviewer 1 Report

  1. In the abstract, sometimes STOFHLA and sometimes S-THOFHLA are described inconsistently. The authors should unified terminology.
  2. The authors say, “The first empirical studies about the validation of HL measurements were published in 2016 in Hungary” Is that true? or does it means that “The first empirical studies in Hungary about the validation of HL measurements were published in 2016 in Hungary [6–7], Please check it.
  3. The introduction states that hypoxia affects health literacy (based on the bibliography). How hypoxia affects illiteracy is not easy to understand. The authors should explain it.
  4. The authors should include the questionnaires used as additional material. (in English).
  5. The authors report that they collected the gender of the participants. Often “gender” is used instead of “Sex” because it seems to be politically correct. Unless the authors have the sexual orientation of the participants (e.g., heterosexual, non-binary, LGTBI), they should use the variable “sex.” According to the World Health Organization, “sex” refers to the biological and physiological characteristics that define men and women, and “gender” refers to the socially constructed roles, behaviors, activities, and attributes that a particular society considers appropriate for men and women. Thus, “sex” is a biological and physical variable, while “gender” is a social, cultural, and psychological variable. See also the report of the National Academy of Medicine. https://genderedinnovations.stanford.edu/Sex%20Specific%20Reporting%20Wizemann%202.pdf
  6. Check the English of the paper to eliminate typing mistakes.
    1. Ej line 184 “cloze items.”
    2. “Our second analysis were the associations of health literacy with (line 238 )” the verb should be “was.”
  7. In my opinion, the authors should organize the article in the following sections: introduction, material and methods, results, and discussion. Within each section, a subsection for each study or a general part can be introduced.
  8. There should be a subsection entitled statistical analysis in the material and methods section. Lines 213-218 should be moved to this subsection.  All the statistical analyses, i.e., Spearman’s rank correlation as a non-parametric test and Mann-Whitney U test (line 239), Dunnett T3 post hoc test should be included in the material and methods. In material and methods section explain that consistency was measured by Cronbach alpha.
  9. In the methodological part, the inclusion and exclusion criteria are indicated, as well as the individuals who decided to participate or not. Still, it is not shown how the patients were selected. For example. Were all patients attending the center offered to participate or were all patients who met the inclusion criteria sought, and 50% were randomly selected, or was it a convenience sampling?

  1. On lines 385 and 387, there are asterisks. subscale rho(69) = -0.242*)subscale rho(69) = -0.242*) I assume that the asterisks mean that the data are statistically significant. In any case, the data should be presented coherently, and if in the rest of the results the p is provided, it should be the same here.

  1. On lines 406-407, there is the following text “(chi-square S-TOFHLA and COPD-Q (69) = 0.142 and chi-square BHLS and COPD-Q (69) 0.403, see Table 3). It isn’t obvious. I suppose that it means Chis Square df(degree of freedom )=69, but I don’t understand what  0,142 stands for either  Chisquare value or P-value.  The authors should use one of these alternatives either (Chi-Square = XXX, df= xxx, P=xxx) or (chi-square P= XXX=.

Any way notation should be consistent used in the whole paper.

  1. In the results, the notation is confused because included between brackets a number that could be the degree of freedom. “Our second analysis were the associations of health literacy with age (negative) and educational level. We used Spearman’s rank correlation as a non-parametric test and Mann-Whitney U test where educational groups were compared. Younger (rho(78) = - 438 p < 0.01) and COPD patients with a high-school or university degree (t-education(2) = -2.115; p = 0.039)”. It is more clear if the authors write rho = -  0.438 p < 0.01) and COPD p (t-education = -2.115; p = 0.039)”. (See above Point)

  1. Table 3 doesn’t provide any interesting information. It should be eliminated or moved to supplementary materials.

Reviewer 2 Report

Dear authors,

Thank you for the opportunity of review your work.

The main issue is interesting. However I consider that the paper has some very important flaws.

GENERAL COMMENTS

The structure of the work is not suitable. The contains are merged in the different sections (introduction, methods, results, discussion) and this text makes that it is very difficult to clearly understand how the work is done.

The sections of an article must be useful to enable the reading of the work, and the confusion alongside the whole paper prevents from a correct understanding of it.

INTRODUCTION

It is too long and confusing. In it some of the issues should be in the discussion section.

In the introduction it seems that the conclusions of the study are already known, that point makes wonder whether the study is really necessary.

ADDING AND HYPOTHESIS

The main aim is quite concrete, but the hypothesis are not clear. Some of them seem to be already proved. The authors include some references in the objective and hypothesis. That is not a good practice.

Some aspects of the methodology are included in this section.

METHODS

Although this section is named as methods it includes information about methodology, results and discussions of two studies. These two studies are not presented and their objectives are not clear.

Data of study 1 are from 2016-2017. Five years is too much time to consider results as a novelty.

The sample size is 80 people with COPD, a very small sample. Neither the size nor the design sampling are detailed.

The results of these study 1 are embedded into the general methods section, and it is merged with the statistical analysis. This statistical analysis is not very detailed. The table 1 shows results about the studies 1 and 2, but the study 2 is not yet presented..

The discussion of this study 1 is also included in the method section.

After study 1 comes the study 2, with its results and its discussion (all this included in the methods section).

The objective of this study 2 is not clear at all. The sample size is 72 patients. There is not sample size nor design or other justifying.

I am sorry to say that perhaps the study can contain interesting data, but its structure makes me impossible to find them out. 

I recommend to the authors to consider a total change in the design of the study.